# Body Weight Counts—Cardioversion with Vernakalant or Ibutilide at the Emergency Department

**DOI:** 10.3390/jcm11175061

**Published:** 2022-08-28

**Authors:** Teresa Lindmayr, Sebastian Schnaubelt, Patrick Sulzgruber, Alexander Simon, Jan Niederdoeckl, Filippo Cacioppo, Nikola Schuetz, Hans Domanovits, Alexander Oskar Spiel

**Affiliations:** 1Department of Emergency Medicine, Medical University of Vienna, 1090 Vienna, Austria; 2Department of Emergency Medicine, Clinic Ottakring, 1160 Vienna, Austria; 3Division of Cardiology, Department of Internal Medicine II, Medical University of Vienna, 1090 Vienna, Austria

**Keywords:** atrial fibrillation, cardioversion, pharmacological cardioversion, body weight

## Abstract

Aim: Medication for the pharmacological cardioversion of atrial fibrillation (AF) and atrial flutter (AFL) is applied either in a fixed dose or adapted to body weight. Individual body weight might be a relevant confounder for anti-arrhythmic treatment success. Therefore, the aim of this study was to elucidate the impact of body weight on pharmacological cardioversion success, comparing weight adapted (Vernakalant) and fixed dose (Ibutilide) pharmacotherapeutic cardioversion regimes. Methods: Within this prospective observational trial, a total of 316 episodes of AF and AFL were enrolled. Patients were stratified in either a Vernakalant (*n* = 181) or Ibutilide (*n* = 135) treatment arm, based on the chosen regime, for direct comparison of treatment efficacy. Results: Conversion to sinus rhythm was achieved in 76.3% of all cases. Of note, there was no difference comparing the Vernakalant and Ibutilide treatment arms (Vernakalant 76.2% vs. Ibutilide 76.3%; *p* = 0.991). Within the whole study population, decreasing conversion rates with increasing body weight (adjusted odds ratio (OR) = 0.69 (0.51–0.94); *p* = 0.018) were observed. An independent effect of body weight within the Ibutilide treatment arm was noted, which remained stable after adjustment for potential confounders (adjusted OR = 0.55 (0.38–0.92), *p* = 0.022. Conclusion: Both, the Vernakalant and Ibutilide treatment arms showed comparable rates of treatment success in pharmacotherapeutic cardioversion of AF and AFL. Of utmost importance, we observed that the fixed dose of Ibutilide—as compared to the weight-adapted dose of Vernakalant—showed a reduced treatment success with increasing body weight.

## 1. Introduction

Atrial fibrillation (AF) and atrial flutter (AFL) are the most common cardiac arrhythmias, with increasing prevalence and incidence worldwide, and are associated with severe complications such as an increased risk for heart failure, ischemic stroke, and overall mortality [1]. Rhythm control should be offered in patients whose daily routine is affected by AF symptoms and in hemodynamically unstable situations. In the latter, electrical cardioversion (EC) is the method of choice and shows much higher conversion rates than pharmacological cardioversion [2,3]. Nevertheless, pharmacological rhythm control is a potent alternative to EC in many circumstances and is preferred by many patients and clinicians [1].

There is a broad spectrum of antiarrhythmic drugs available for rhythm control in patients with AF [1]. Most drugs are administered in a fixed dose, however, some drugs, e.g., the recently marketed relatively atrial selective antiarrhythmic agent Vernakalant, are given in a weight-adapted dose.

Interestingly, there are various links between obesity and AF prevalence and treatment success: according to the Framingham Heart Study a one unit increase in Body Mass Index (BMI) is correlated with a 4%–5% higher risk of developing AF [4]. Obesity affects AF and AFL development and severity in several different ways: On the one hand, cardiac remodeling—such as replacement of myocardial tissue by adipocytes—changes myocardial electrical conduction and, therefore, predisposes for arrhythmia. On the other hand, chronic inflammation of myocytes leading to fibrosis was observed, again negatively influencing myocardial conduction properties [1]. However, the literature is scarce concerning the question if obesity and body weight affect pharmacological cardioversion success.

Both Vernakalant and Ibutilide are intravenously applied class III antiarrhythmics with comparable pharmacokinetic properties and dosing regimens, with up to two short infusions over 10 min. For Vernakalant, the pattern of usage provides 3 mg/kg over 10 min as an infusion for the initial dose and 2 mg/kg over 10 min in case of subsisting arrhythmia 15 min after the first application [5,6,7].

In an RCT comparing Vernakalant and Ibutilide, Simon et al. [8] observed higher conversion rates in the Vernakalant group. However, AFL episodes were excluded due to lacking effect of Vernakalant on this arrhythmia—especially when compared to Ibutilide [9]. Therefore, our goal was to contrast the overall cardioversion success of Ibutilde vs. Vernakalant in real-life usage at the emergency department (ED) in AF and AFL patients.

Consequently, the study aim was to compare, depending on patients’ body weight, the cardioversion success rates between variable Vernakalant doses and fixed Ibutilide doses. Besides the main focus (cardioversion success depending on body weight), this study also discusses some co-variates, such as effect on QTc time and influence of comorbidities on treatment success.

## 2. Materials and Methods

Study data were extracted from the “Atrial Fibrillation Registry” of the Department of Emergency Medicine, Vienna General Hospital, Austria. This registry is an electronic database containing every AF and AFL episode that was recorded at the department. Moreover, it includes information such as therapy strategies, adverse events, or patients’ histories. The registry is filled by specially trained study personnel, and all entries are supervised and double-checked by the study physicians. AF and AFL episodes (including adverse events) have continuously been enrolled prospectively, since 2012, strictly adhering to recent data policies. All patients were consecutively pseudonymized, and the study protocol was approved by the Ethics Committee of the Medical University of Vienna (updated EC No.: 2017/1938).

### 2.1. Patient Enrollment and Data Acquisition

From January 2013 until December 2016, a total of 316 symptomatic AF (*n* = 248) and AFL (*n* = 68) episodes were included. Of these, 181 were treated with Vernakalant and 135 with Ibutilide, for acute restoration of sinus rhythm (SR) at the ED. Choice of the drug in the respective cases was at the discretion of the attending physician, who was not involved in the study design. It was based on personal preferences, indications, and contraindications of the two drugs, always in line with current guidelines [1]. Exclusion criteria consisted of episodes with contraindications to pharmacological cardioversion (e.g., hemodynamic instability) or contraindications to Ibutilide or Vernakalant (e.g., known allergy against either drug, severe heart failure, acute coronary syndrome within the past 30 days, atrioventricular block grade II or III, QTc time > 500 ms, sick sinus syndrome, Wolff–Parkinson–White syndrome, and pretreatment with other class I or III antiarrhythmic drugs) [10,11]. In addition, episodes of permanent AF were excluded because rhythm control is not clearly indicated in this population [1].

In accordance with current AF guidelines [1], left atrium thrombus exclusion by imaging (transesophageal echocardiography or CT scan of left atrium auricle) was performed in patients without previous oral anticoagulation and with AF symptoms lasting longer than 48 h before cardioversion. Before treatment, potassium levels were measured in every patient. If potassium value was <4.2 mmol/L, a pre-fabricated electrolyte infusion (Elozell spezial, Fresenius Kabi-24 mmol potassium, and 6 mmol magnesium) was administered prior to cardioversion.

Patients received up to two short infusions of Ibutilide (first and second infusion: 1 mg, if adjusted weight > 60 kg; <60 kg: 0.01 mg/kg) or Vernakalant (first infusion: 3 mg/kg body weight; second infusion: 2 mg/kg adjusted body weight) with an in-between interval of 10 min (Ibutilde) or 15 min (Vernakalant), in accordance with the recommendation of the manufacturer. During (and until 4 h after) treatment, patients continuously received oxygen saturation-, blood pressure-, and ECG-monitoring. Relevant medical history was recorded, especially presence of arterial hypertension, coronary heart disease, prior stroke, vascular disease, congestive heart failure, diabetes mellitus, and hyperlipidemia. In addition, everyday medication (e.g., anticoagulation, ß-blocker-therapy, ACE inhibitors, angiotensin receptor blockers and statins) was recorded in all patients in the Atrial Fibrillation Registry.

### 2.2. Patient Follow-Up and Endpoint Analysis

The primary study endpoint was defined as “restoration of SR using either Vernakalant or Ibutilide”, within 2 h after drug administration. In case of conversion during intravenous treatment with the study medication, we stopped the infusion with Ibutilide, whereas the entire dose of Vernakalant was applied following the specification of both drugs. Prolongation of QTc time was defined as secondary endpoint.

### 2.3. Statistical Analysis

For statistical analysis, logistic regression to evaluate the impact of patients’ body weight and BMI on cardioversion success was used. The same model was used for the impact of patients’ body heights. The results were adjusted for gender and age, and the risk increase for continuous variables refer to an increase peri one standard deviation (1-SD) (Table 1). Discrete data are presented as counts and percentages and were analyzed using testing for linear association among groups (Mantel–Haenszel chi-squared test). Continuous variables are shown as medians and their respective interquartile range (IQR) and were compared using Mann–Whitney U test.

## 3. Results

### 3.1. Study Population

The median age of included patients was 64 (50–73) years. There was a trend in Vernakalant recipients being slightly younger, however, this was not statistically significant (mean age distribution in the Ibutilide group 67 (52–75) years vs. Vernakalant group 62 (49–70) years). In total, 195 (61.7%) of the included patients were males (64.1% Vernakalant, 58.5% Ibutilide). In regard to body height, body weight, and BMI, patient characteristics were found to be balanced between treatment strategies (Table 2).

### 3.2. Efficacy and Effect on QTc

The overall pharmacological cardioversion success among all 316 participants was 76.3% (241 patients), with nearly identical cardioversion success in both treatment arms: Vernakalant: 76.2% (138 of 181 patients) and Ibutilide: 76.3 (103 of 135 patients) (Table 2).

Ibutilide is known to be more effective in AFL than in AF [8,9,10]. Therefore, we analyzed conversion success considering the first detected ECG rhythm in both subpopulations. In the Ibutilide group, we observed higher conversion rates in AFL than in AF, as expected (AFL 59 out of 67 participants, 88.1%; AF 44 out of 68, 64.7%). One patient with AFL was mistakenly treated with Vernakalant. As expected, they did not convert to sinus rhythm.

Vernakalant did not have negative (prolonging) effects on the patients’ QTc-interval (before cardioversion: treatment success 443.0 (421.0–463.0) ms; no treatment success 448.5 (421.8–483.5) ms, *p* = 0.380; after cardioversion: treatment success 430.5 (406.3–456.8) ms; no treatment success 439.0 (414.5–476.3) ms, *p* = 0.292). On the contrary, we did observe (but it was not statistically significant) QTc-shortening after administration of Vernakalant. Ibutilide patients, on the other hand, showed a (not significant) trend to remarkable prolongation of the QTc-interval, especially in participants without conversion success (before cardioversion: treatment success 455.0 (428.5–472.8) ms; no treatment success 456.5 (438.3–488.0) ms, *p* = 0.514; after cardioversion: treatment success 466.0 (441.0–497.0) ms; no treatment success 500.0 (435.5–518.0) ms, *p* = 0.058) (Table 3, Table 4 and Table 5).

None of the patients with known heart failure (*n* = 6) could be cardioverted successfully in the Vernakalant group (*p* = 0.001). In the Ibutilide subpopulation, 62.5% of patients who suffered from arterial hypertension did not convert to sinus rhythm (*p* = 0.032). (Table 3).

### 3.3. Correlation of Body Weight and Conversion Success

A median body weight of 80 (71–91) kg was found in the overall population. There was no significant difference concerning mean body weight in the two groups (Vernakalant 83 (73–92) kg; Ibutilide 80 (70–91)). BMI accounted for a median of 26.2 (23.8–29.2) kg/m^2^ in the study (Ibutilide: 26.0 (23.0–29.4) kg/m^2^, Vernakalant: 26.3 (24.1–29.1) kg/m^2^) (Table 2).

We observed significantly decreasing conversion rates with increasing body weight (adjusted odds ratio (OR) = 0.69 (0.51–0.94); *p* = 0.018) (Table 1) in the whole study population (Appendix A). There was no significant impact of BMI or body height on conversion success.

Of utmost importance, an independent effect of body weight within the Ibutilide treatment arm was noted, which remained stable after adjustment for age and gender (adjusted OR = 0.55 (0.38–0.92), *p* = 0.022) (Table 5). No such effect was observed in patients treated with Vernakalant (adjusted OR = 0.85 (0.58–1.24)); *p* = 0.403; (Table 5).

## 4. Discussion

The present study represents—to the best of our knowledge—the largest investigation on weight-dependence of pharmacological cardioversion success in recent–onset atrial fibrillation at the emergency department. The provided data demonstrates that conversion success of Ibutilide, which is given in a fixed dose in patients with body weight greater then 60 kg, decreases with increasing body weight. This effect was not observed with Vernakalant, which is applied in a weight-adjusted fashion at all weights. Moreover, our data shows a (statistically not significant) trend for lower conversion rates in patients > 90 kg treated with Vernakalant too. This is potentially caused by the limitation of Vernakalant for patients with a body weight higher than 113 kg. The maximum approved dose of Vernakalant is 339 mg for the first administration and 226 mg for the second administration [5]. Hence, patients with higher body weight should not receive weight-adapted treatment. Therefore, electrical cardioversion may be a better option for these patients, if there are no contraindications against the procedure.

To our knowledge, this is the first study to show that obesity is a negative predictor for cardioversion success in patients treated with Ibutilide. A possible explanation, in our opinion, might be a fixed, non-weight-adapted dose, leading to lower concentration of the active agent at the site of action. However, obesity itself is linked to AF prevalence and associated with treatment success: according to the Framingham Heart Study, a one unit increase in BMI correlates with a 4%–5% higher risk of developing AF [4].

Nevertheless, the question if higher doses of Ibutilide are necessary in certain patients arises. However, referring to the phase II trial [7], higher plasma levels of Ibutilide are associated with a significant prolongation of the QTc-interval, without an increase in conversion success. Therefore, higher doses of Ibutilide might be predisposing for a higher number of complications such as proarrhythmia, but, in our data, no significant impact on QTc-interval could be observed.

Of note, we found high conversion rates in both groups compared to other studies that analyzed the effectiveness of the drugs used for cardioversion [8,12]. The ED setting might be a reason for this controversy, since ED populations show a higher rate of new-onset AF and, therefore, shorter time intervals from AF onset to drug administration than a general AF population [9,13]. Unfortunately, our database lacks sufficient information about the mean duration from arrhythmia onset to time of conversion.

Viktorsdottir et al. [14], analyzing the cardioversion success of Ibutilide in an ED in 2006, observed similar conversion rates as in our study, suggesting internationally comparable data.

The high-rate of treatment failure in Ibutilide-treated patients with arterial hypertension might be linked to hypertension-induced structural remodeling or fibrosis that causes myocardial conduction disturbances, which is predisposing to AF development [15,16] and could, furthermore, lead to a lower rate of cardioversion success. A theory why this effect occurs in patients treated with Ibutilide but not in participants treated with Vernakalant may be the different localization of the responding potassium channels: I_Kur_ channels (Vernakalant) are located superficially in intercalated discs, whereas the potassium channels that are affected by Ibutilde are located deeper in the myocardial tissue [17,18,19]. As a consequence, it may be more difficult to reach the I_Kr_ channels when being affected by left-ventricular hypertrophy, which is caused by arterial hypertension [20].

Besides this, no one in the Vernakalant group with known heart failure showed successful restoration of SR—an effect that proved to be statistically significant, but could as well be due to chance, as the sample size in this group is rather small. The available data [8] assessed the performance of Vernakalant in patients with mild heart failure (NYHA I or II) and ischemic heart disease [4]. In front of the background of the current guidelines recommending Vernakalant for use in this special patient subgroup and our discouraging results, further research with a larger collective of different grades of heart failure is needed.

Moreover, a significant prolongation of the QTc interval after an attempt at cardioversion was observed, which was most pronounced in the Ibutilide subgroup without conversion success. This is probably caused by the higher plasma concentration of Ibutilide in the group without treatment success, because these patients received the second dose (2 milligrams) of the agent in contrast to those Ibutilide patients successfully cardioverted after the first dose. It can be assumed that this effect cannot be found in Vernakalant patients because it is a more atria-selective drug, with only moderate effects on QTc intervals [8]. Another practical aspect, which might also play a role in choosing fixed dose Ibutilide, is that no complex calculations and dilutions of the appropriate drug dose are necessary.

As is well-known, Ibutilide is more effective in AFL than in AF [4]. According to Simon A. et al. [8], who compared Vernakalant and Ibutilide, including only episodes of AF, and described a higher rate of restoration of SR in the Vernakalant group, we could prove that Ibutilide is equipotent to Vernakalant, when analyzing together AF and AFL patients in a real-life ED setting.

Following our findings, paying attention to patients’ body weights could be a useful tool for clinical decision making, particularly in obese AF patients. Therefore, weight-adapted medication or electrical cardioversion should be considered for those patient groups. In addition, Ibutilide is an effective and safe agent for pharmacological cardioversion in low- and normal-weight AF and, especially, AFL patients.

### Limitations

The study has several limitations. The rather small sample size, the single-center perspective, and the non-randomized design can only give an impulse for further research and cannot translate into general treatment recommendations. Aside from that, time of onset to cardioversion has a known influence on the treatment success of rhythm control [21]. Due to a lack of data, it was not possible to take AF duration into account adequately. Furthermore, because of the slight difference concerning the indication (Vernakalant for AF and Ibutilide for AF and AFL), a straight comparison of both drugs is problematic. However, the prospectively collected data involving all patients treated for AF at the Department of Emergency Medicine allows us to overcome the most relevant bias.

## 5. Conclusions

In a real-life setting at the emergency department, Vernakalant and Ibutilide are overall equally effective in converting atrial fibrillation and atrial flutter to sinus rhythm. The fixed dose of Ibutilide—as compared to the weight-adapted dose of Vernakalant—showed a reduced treatment success with increasing body weight. As a consequence, these results might have an influence on anti-arrhythmic treatment selection in overweight patients undergoing pharmacological cardioversion.

## Figures and Tables

**Table 1 jcm-11-05061-t001:** Impact of BMI, height, and body weight on cardioversion success.

	Crude OR (95% CI)	*p*-Value	* Adjusted OR (95% CI)	*p*-Value
BMI	0.77 (0.58–0.99)	0.048	0.77 (0.61–1.01)	0.059
Height	0.94 (0.78–1.15)	0.150	0.89 (0.75–1.15)	0.203
Body weight	0.74 (0.57–0.97)	**0.029**	0.69 (0.51–0.94)	**0.018**

BMI = Body Mass Index, OR = odds ratio, CI = confidence interval. * adjusted to sex and age. Bold was used for statistically significant results (*p*-value < 0.05). Results greater than 0.05 are not significant.

**Table 2 jcm-11-05061-t002:** Baseline characteristics for the total population.

	Total	Vernakalant	Ibutilide	*p*-Value
Age, years (IQR)	**64** (50–73)	**62** (49–70)	**67** (52–75)	**0.003**
Male sex, *n* (%)	**195** (61.7)	**116** (64.1)	**79** (58.5)	0.314
Cardioversion success, *n* (%)	**241** (76.3)	**138** (76.2)	**103** (76.3)	0.991
Height, cm (IQR)	**175** (168–183)	**176** (168–184)	**175** (165–182)	0.159
Weight, kg (IQR)	**80** (71–91)	**83** (73–92)	**80** (70–91)	0.040
BMI, kg/m^2^ (IQR)	**26.2** (23.8–29.2)	**26.3** (24.1–29.1)	**26.0** (23.0–29.4)	0.115
First detected rhythm, *n* (%)				**0.001**
Atrial fibrillation	**248** (78.5)	**180** (99.4)	**67** (49.6)	
Atrial flutter	**68** (21.5)	**1** (0.6)	**68** (50.4)	
Classification, *n* (%)				0.441
Paroxysmal/persistent	**231** (73.1)	**133** (73.4)	**98** (72.6)	
New onset	**27** (8.5)	**24** (13.3)	**3** (2.2)	
Not known	**58** (18.4)	**24** (13.3)	**34** (25.2)	
Comorbidities, *n* (%)				
Arterial hypertension	**161** (50.9)	**99** (54.7)	**62** (45.9)	0.123
Heart failure	**16** (5.1)	**6** (3.3)	**10** (7.4)	0.101
Coronary heart disease	**23** (7.3)	**19** (10.5)	**4** (3.0)	0.011
Prior stroke/TIA	**23** (7.3)	**14** (7.7)	**9** (6.7)	0.718
Vascular disease	**26** (8.2)	**14** (7.7)	**12** (8.9)	0.712
Diabetes mellitus	**19** (6.0)	**13** (7.2)	**6** (4.4)	0.312
Hyperlipidemia	**77** (24.4)	**51** (28.2)	**26** (19.3)	0.068
Long-term medication, *n* (%)				
Beta blocker	**145** (45.9)	**88** (48.6)	**57** (42.2)	0.260
Amiodarone	**15** (4.7)	**11** (6.1)	**4** (3.0)	0.198
Digitalis	**5** (1.6)	**1** (0.6)	**4** (3.0)	0.090
Flecanide	**12** (3.8)	**4** (2.2)	**8** (5.9)	0.088
Dronedarone	**2** (0.6)	**1** (0.6)	**1** (0.7)	0.835
Propafenone	**1** (0.3)	**1** (0.6)	**0** (0.0)	0.388
Ca-channel inhibitors	**42** (13.3)	**25** (13.8)	**17** (12.6)	0.752
Alpha blockers	**11** (3.5)	**9** (5.0)	**2** (1.5)	0.095
ACE inhibitors	**45** (14.2)	**31** (17.1)	**14** (10.4)	0.090
AT_2_ inhibitors	**90** (28.5)	**54** (29.5)	**36** (26.7)	0.538
Diuretics	**64** (20.3)	**31** (17.1)	**33** (24.4)	0.110
Lipid-lowering agents	**62** (19.6)	**46** (25.4)	**16** (11.9)	**0.003**
P2Y_12_ antagonists	**14** (4.4)	**11** (6.1)	**3** (2.2)	0.100
Acetylsalicylic acid	**52** (16.5)	**41** (22.7)	**11** (8.1)	**0.001**
Phenprocoumon	**45** (14.2)	**18** (9.9)	**27** (20)	**0.012**
NOACs	**78** (24.7)	**44** (24.3)	**34** (25.2)	0.866
LMWH	**4** (1.3)	**4** (2.2)	**0** (0.0)	0.083

Values = patients, *n* (%), IQR = interquartile range, BMI = Body Mass Index, TIA = transient ischemic attack, Ca = calcium, ACE = angiotensin-converting enzyme, AT_2_ = angiotensin receptor II, NOACs = novel oral anticoagulants, LMWH = low molecular weight heparin, lipid-lowering agents including statins, fibrates, and ezetimibe. Bold was used for statistically significant results (*p*-value < 0.05). Results greater than 0.05 are not significant. Bold was used for statistically significant results (*p*-value < 0.05). Results greater than 0.05 are not significant.

**Table 3 jcm-11-05061-t003:** Comorbidities and conversion success.

**Vernakalant Population**			
	**Treatment Success**	**No Treatment Success**	** *p* ** **-Value**
Arterial hypertension, *n* (%)	76 (55.1)	23 (53.5)	0.856
Heart failure, *n* (%)	0 (0.0)	6 (14.0)	**0.001**
Coronary heart disease, *n* (%)	13 (9.4)	6 (14.0)	0.398
Prior stroke/TIA, *n* (%)	10 (7.2)	4 (9.3)	0.660
Vascular disease, *n* (%)	13 (9.4)	1 (2.3)	0.129
Diabetes mellitus, *n* (%)	9 (6.5)	4 (9.3)	0.539
Hyperlipidemia, *n* (%)	39 (28.3)	12 (27.9)	0.964
CHA_2_DS_2_-VASC Score (SD)	1.33 (1.26)	1.65 (1.27)	0.097
Adverse events, *n* (%)	10 (7.2)	4 (9.3)	0.660
Type of adverse events, *n* (%)			0.710
Atrial flutter	5 (3.6)	4 (9.3)	
Bradycardia	5 (3.6)	0 (0.0)	
Torsade de Pointes arrhythmia	0 (0.0)	0 (0.0)	
None	126 (91.3)	35 (81.4)	
QTc-interval pre-cardioversion (ms), (IQR)	443.0 (421.0–463.0)	448.5 (421.8–483.5)	0.380
QTc-interval post-cardioversion (ms), (IQR)	430.5 (406.3–456.8)	439.0 (414.5–476.3)	0.292
**Ibutilide Population**			
	**Treatment Success**	**No Treatment Success**	** *p* ** **-value**
Arterial hypertension, *n* (%)	42 (40.8)	20 (62.5)	**0.032**
Heart failure, *n* (%)	10 (9.7)	0 (0.0)	0.068
Coronary heart disease, *n* (%)	3 (2.9)	1 (3.1)	0.951
Prior stroke/TIA, *n* (%)	8 (7.8)	1 (3.1)	0.360
Vascular disease, *n* (%)	8 (7.8)	4 (12.5)	0.413
Diabetes mellitus, *n* (%)	4 (3.9)	2 (6.3)	0.572
Hyperlipidemia, *n* (%)	17 (16.5)	9 (28.1)	0.147
CHA_2_DS_2_-VASC Score (SD)	1.39 (1.22)	1.41 (1.01)	0.601
Adverse events, *n* (%)	18 (17.4)	1 (3.1)	**0.042**
Type of adverse events, *n* (%)			**0.037**
Atrial flutter	4 (3.9)	1 (3.1)	
Bradycardia	13 (12.6)	0 (0.0)	
Torsade de Pointes arrhythmia	1 (1.0)	0 (0.0)	
None	82 (79.6)	29 (90.6)	
QTc-interval pre-cardioversion (ms), (IQR)	455.0 (428.5–472.8)	456.5 (438.3–488.0)	0.514
QTc-interval post-cardioversion (ms), (IQR)	466.0 (441.0–497.0)	500.0 (435.5–518.0)	0.058

Values = patients, *n* (%), SD = standard deviation, IQR = interquartile range, TIA = transient ischemic attack, ms = milliseconds. CHA_2_DS_2_-Vasc score including congestive heart failure (1 point), hypertension (1 point), age (>75 years 2 points), diabetes mellitus (1 point), stroke/TIA (2 points) vascular disease (1 point), age (65–74 years 1 point), sex (female 1 point). Bold was used for statistically significant results (*p*-value < 0.05). Results greater than 0.05 are not significant.

**Table 4 jcm-11-05061-t004:** Results for the total population.

	Total	Vernakalant	Ibutilide	*p*-Value
Cardioversion success, *n* (%)	241 (76.3)	138 (76.2)	103 (76.3)	0.991
Adverse events, *n* (%)	44 (13.9)	20 (11)	24 (17.8)	0.069
Type of adverse events, *n* (%)				0.059
Atrial fibrillation	5 (1.5)	0 (0.0)	5 (3.7)	
Atrial flutter	9 (2.8)	9 (5.0)	0 (0.0)	
Bradycardia	18 (5.7)	5 (2.8)	13 (9.6)	
Torsade de Pointes arrhythmia	1 (0.3)	0 (0.0)	1 (0.7)	
None	272 (86.1)	161 (89.0)	111 (82.2)	
Cardioversion success in detail, *n* (%)				0.213
No SR	75 (23.7)	43 (23.8)	32 (23.7)	
SR after 1st application	165 (52.2)	102 (56.4)	63 (46.7)	
SR after 2nd application	76 (24.1)	36 (19.9)	40 (29.6)	
CHA_2_DS_2_-VASC score (standard deviation)	1.40 (1.23)	1.40 (1.27)	1.39 (1.17)	0.872
QTc-interval pre-cardioversion (ms), (IQR)	450.0 (424.5–472.0)	445.0 (421.5–465.5)	455.5 (434.5–474.5)	0.061
QTc-interval post-cardioversion (ms), (IQR)	450.5 (421.0–486.0)	433.5 (408.0–459.3)	470.5 (441.0–500.3)	**0.001**

Values = patients, *n* (%), IQR = interquartile range, ms = milliseconds; CHA_2_DS_2_-Vasc score including congestive heart failure (1 point), hypertension (1 point), age (>75 years 2 points), diabetes mellitus. Bold was used for statistically significant results (*p*-value <0.05). Results greater than 0.05 are not significant.

**Table 5 jcm-11-05061-t005:** Impact of body weight on cardioversion success for both subpopulations.

	Crude OR (95% CI)	*p*-Value	* Adjusted OR (95% CI)	*p*-Value
Vernakalant	0.87 (0.61–1.23)	0.429	0.85 (0.58–1.24)	0.403
Ibutilide	0.62 (0.38–0.94)	0.024	0.55 (0.38–0.92)	**0.022**

OR = odds ratio, CI = confidence interval. * adjusted to sex and age. Bold was used for statistically significant results (*p*-value < 0.05). Results greater than 0.05 are not significant.

## Data Availability

The datasets acquired and/or analyzed during the current study are available from the corresponding author on reasonable request.

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
