# Peer review of "Body Weight Counts—Cardioversion with Vernakalant or Ibutilide at the Emergency Department"

_jcm, 2022, doi:10.3390/jcm11175061_

Round 1
Reviewer 1 Report
Dear authors,
thank you very much for the possibility to review this paper.
A-fib and a-flutter are common problems in the ED. Updates to management of these entities are an important contribution to the field of cardiological emergency medicine.
I have carefully reviewed and would like to add some remarks:
ll. 84 ff. was prolonged QT time a contraindication?
ll. 84 ff. were electrolytes measured before treatment? -> Were electrolyte disturbance, especially hypo-/hyperkalemia treated as a contraindication or corrected before treatment?
l. 89 you state that „pretreatment with other class I or III antiarrhythmic drugs“ is an exclusion criterion. However, in table 2 you state that patients were pre-treated with amiodaron and dronedarone (class III) / Flecainide and Propafenone (Ic).
ll. 89/90 you state that episodes of permanent AF were excluded because rhythm control is not clearly indicated in this population . However, in table 2 you state that in some patients the onset of A-fib/A-flutter was not known.
l. 92 thrombus exclusion by imaging (TTE or TEE?)
l.95 ff. was treatment stopped if cardioversion was achieved during treatment?
ll. 129 ff.: „Marcoumar“ (brand name)-> Phenprocoumon; Acetylasilcylic acid (misspelled)-> acetylsalicylic acid
Was ideal body weight or adjusted body weight taken into account?
Were patients included in this study who arrived to the ED with complaints fixed to A-Fib or A-Flutter or also patients in which accidentally A-Fib/-flutter was diagnosed via monitoring or 12-lead-ECG with diffeent
Kind regards
Author Response
Dear Reviewer,
We would like to thank you for your comments which enabled us to substantially improve our manuscript.
All changes are explained point by point in a response to the reviewers and editors or below.
We sincerely hope this revised manuscript is suitable for publication Journal of Clinical Medicine and thank you for your consideration.
84 ff. was prolonged QT time a contraindication?
Authors’ response: Participants with QTc >500ms were excluded, we added this information in the manuscript (page 2, line 91).
84 ff. were electrolytes measured before treatment? -> Were electrolyte disturbance, especially hypo-/hyperkalemia treated as a contraindication or corrected before treatment?
Authors’ response: Following our local protocols, electrolyte levels (and in particular, potassium) are measured in every patient before any treatment is provided. If potassium values were not in the high normal range (>4.2 mmol/L), patients received a pre-fabricated electrolyte infusion (Elozell ‘spezial’, 250 mL, Fresenius Kabi, Austria) with 24 mmol of potassium and 6 mmol of magnesium prior to cardioversion. We added this information in the manuscript (page 2, line 91).
89 you state that „pretreatment with other class I or III antiarrhythmic drugs“ is an exclusion criterion. However, in table 2 you state that patients were pre-treated with amiodaron and dronedarone (class III) / Flecainide and Propafenone (Ic).
Authors’ response: Here, chronic medication was meant, and patients could be on long-term medication without being excluded, as illustrated in Table 2. Of importance, no intravenous or oral antiarrhythmic treatment (class I or III drugs) with exception of the study medication was used to restore sinus rhythm.
89/90 you state that episodes of permanent AF were excluded because rhythm control is not clearly indicated in this population. However, in table 2 you state that in some patients the onset of A-fib/A-flutter was not known.
Authors’ response: We defined permanent AF if the arrhythmia was already classified as permanent prior to presentation to our department. If a patient was diagnosed with AF and had episodes terminated by medical or electrical intervention, or converted to sinus rhythm spontaneously and the onset of the index episode was not known, we classified this as persistent AF and therefore patients could be included.
92 thrombus exclusion by imaging (TTE or TEE?)
Authors’ response: We made thrombus exclusion either by TEE or atrial auricle CT-scan. We added this information in the manuscript (page 3, line 96).
l.95 ff. was treatment stopped if cardioversion was achieved during treatment?
Authors’ response: Following the specifications of the study medication, Ibutilide was stopped when sinus rhythm is restored, whereas participants in the Vernakalant group received the whole infusion completely despite of treatment success during application. We added this information in the manuscript (page 3, lines114).
129 ff.: „Marcoumar“ (brand name)-> Phenprocoumon; Acetylasilcylic acid (misspelled)-> acetylsalicylic acid
Authors’ response: Thank you for taking notice, we corrected both in the manuscript.
Was ideal body weight or adjusted body weight taken into account?
Authors’ response: We took adjusted body weight into account. This was added page 3, line 103 and 104
Were patients included in this study who arrived to the ED with complaints fixed to A-Fib or A-Flutter or also patients in which accidentally A-Fib/-flutter was diagnosed via monitoring or 12-lead-ECG with different
Authors’ response: The patients who underwent cardioversion consulted the ED because of symptoms due to AF or AFL (e.g., palpitations, shortness of breath, vertigo, etc.). We changed the wording in the manuscript (page 2, line 82).
Yours sincerely,
Teresa Lindmayr, MD
Reviewer 2 Report
Thank you for the opportunity to review this manuscript. The authors perform a prospective, observational study to explore the effect of body weight on the success rate of AF cardioversion. Ibutilide showed significant lower conversion rates with increasing body weight,
whereas this effect could not be observed in the vernakalant group. In general, the paper at times strays from its primary objective and discusses other potential effect modifiers. If going to extend the discussion in this way, the authors should mention this in the introduction, methods, and statistical analysis.
line 35- Change to “Rhythm control should be OFFERED to patients whose daily routine is affected by AF symptoms, … “
The decision to cardiovert in stable AF should be shared between clinician and patient.
line 36- clarify that electrocardioversion is the “method of choice” bc it is more effective overall and consider citing:
doi: 10.1016/S0140-6736(19)32994-0
DOI: 10.1007/s43678-020-00067-7
line 84- How did the authors confirm that the selection of drugs were “always in line with current guidelines.”
line 102- This is not relevant to the study objective and can be removed: “Therefrom the CHA2DS2-VASc-Score was calculated in all patients.”
line 132- “effect on QTc” was not mentioned as a pre-specified, secondary outcome.
line 178- List the confounders, gender and age, instead of writing "potential". There are other potential confounders that were not included in the adjustment such as AF duration and LA size
Discussion:
line 185- change to : ….given in a fixed dose in patients with body weight greater than 60kg.
line 187- change to: … which is applied in a weight-adjusted fashion AT ALL WEIGHTS
line 188- “severe adiposity” was never defined
lines 203-205- much of the discussion appears the same as the introduction. Does it need to be repeated?
lines 205-210- These factors are not relevant to the study objective. How does this relate to the proposition that weight may be associated with cardioversion success?
line 215- The available evidence contradicts this. After repeated doses of ibutilide the mean duration was 500ms which may be the critical threshold for pro-arrhythmia ((doi: 10.1111/acem.13702)
line 217- conversion rates of which drugs?
Another direct-comparison trial that should be cited: PMID: 30455558
line 221- AF duration is likely a significant covariate and warrants further discussion as a limitation
line 243- Change to: ….a significant prolongation of the QTc-interval after ATTEMPT AT cardioversion was observed
line 250- fixed-dose ibutilde?
Limitations:
AF duration is likely to be a significant covariate and warrants further discussion as a major limitation
Charts and tables are appropriate on the whole.
Table 5:
Spelling of “ibutilide” is incorrect
Author Response
Dear Reviewer,
We would like to thank you for your comments which enabled us to substantially improve our manuscript.
All changes are explained point by point in a response to the reviewers and editors or below.
We sincerely hope this revised manuscript is suitable for publication Journal of Clinical Medicine and thank you for your consideration.
line 35- Change to “Rhythm control should be OFFERED to patients whose daily routine is affected by AF symptoms, … “ The decision to cardiovert in stable AF should be shared between clinician and patient.
Authors’ response: Thank you, we adapted this according to your suggestion (page 1, line 35).
line 36- clarify that electrocardioversion is the “method of choice” bc it is more effective overall and consider citing: doi: 10.1016/S0140-6736(19)32994-0, DOI: 10.1007/s43678-020-00067-7
Authors’ response: We adapted this in the Introduction (page, 1 lines 37/38).
line 84- How did the authors confirm that the selection of drugs were “always in line with current guidelines.”:
Authors’ response: The drugs available and administered for cardioversion of AF and AFL in our department are documented in the local registry, which allows tracking of use. We double checked this when we acquired the data from the registry.
line 102- This is not relevant to the study objective and can be removed: “Therefrom the CHA2DS2-VASc-Score was calculated in all patients.”
Authors’ response: Thank you, we removed this part following your suggestion.
line 132- “effect on QTc” was not mentioned as a pre-specified, secondary outcome.
Authors' response: We agree and now added the prolongation of QTc as a secondary endpoint to the methods section (page 3, lines 117).
line 178- List the confounders, gender and age, instead of writing "potential". There are other potential confounders that were not included in the adjustment such as AF duration and LA size
Authors’ response: Thank you, we adapted this in the manuscript (page 7, line 193).
line 185- change to : ….given in a fixed dose in patients with body weight greater than 60kg.
Authors’ response: Thank, you we adapted this according to your suggestion (page 7, line 200).
line 187- change to: … which is applied in a weight-adjusted fashion AT ALL WEIGHTS
Authors’ response: Thank, you we adapted this according to your suggestion (page 7, line 202).
line 188- “severe adiposity” was never defined
Authors’ response: We observed this effect especially in patient groups >90kg, which we defined as severely adipose. We changed the wording in the manuscript (page 7, line 204).
lines 203-205- much of the discussion appears the same as the introduction. Does it need to be repeated?
Authors’ response: We agree and thus removed parts of the mentioned sections.
lines 205-210- These factors are not relevant to the study objective. How does this relate to the proposition that weight may be associated with cardioversion success?
Authors’ response: We approve your objection and removed this part entirely.
line 217- conversion rates of which drugs? Another direct-comparison trial that should be cited: PMID: 30455558
Authors’ response: We corrected the wording and cited PMID: 30455558 as this article is indeed of importance to our findings (page 7, line 222).
line 221- AF duration is likely a significant covariate and warrants further discussion as a limitation:
Authors’ response: Thank you for pointing out this limitation. We extended the Limitations section following your recommendation (page 8, line 271).
line 243- Change to: ….a significant prolongation of the QTc-interval after ATTEMPT AT cardioversion was observed
Authors’ response: Thank, you we adapted this according to your suggestion (page 8, line 249).
line 250- fixed-dose ibutilde?
Authors’ response: Thank, you we adapted this according to your suggestion (page 8, line 256).
Table 5: Spelling of “ibutilide” is incorrect
Authors’ response: Thank, you we adapted this according to your suggestion
line 215- The available evidence contradicts this. After repeated doses of ibutilide the mean duration was 500ms which may be the critical threshold for pro-arrhythmia ((doi: 10.1111/acem.13702)
Authors’ response: Could you please specify or rephrase your annotation – thank you!
Yours sincerely,
Teresa Lindmayr, MD
Reviewer 3 Report
Interesting, simple article.
Minor remarks
- highlight the total number of patients in table 2
- Reasonable results
- Statistical methodology well done
- I did not understand in this observational study who or what chose the drug for the patient (ibutilide vs. Vernakalant)
Author Response
Dear Reviewer,
We would like to thank you for your comments which enabled us to substantially improve our manuscript.
All changes are explained point by point in a response to the reviewers and editors or below.
We sincerely hope this revised manuscript is suitable for publication Journal of Clinical Medicine and thank you for your consideration.
highlight the total number of patients in table 2
Authors’ response: Thank you, we changed this according to your suggestion (page 4)
- I did not understand in this observational study who or what chose the drug for the patient (ibutilide vs. Vernakalant)
Authors’ response: The choice of drug in the respective cases was at the discretion of the attending physician, who was not involved in the study design. It was based on personal preferences, indications and contraindications of the two drugs, and / or the patients’ choice in a shared decision process, and it was always in line with current guidelines.
Yours sincerely,
Teresa Lindmayr, MD